# A systematic review and meta-analysis of human and zoonotic dog soil-transmitted helminth infections in Australian Indigenous communities

Cameron Raw 📧 *, Rebecca J. Traub, Patsy A. Zendejas-Heredia, Mark Stevenson, Anke Wiethoelter

Faculty of Veterinary and Agricultural Sciences, University of Melbourne, Parkville, Victoria, Australia

* cameron.raw@unimelb.edu.au

**Data Availability Statement:** All relevant data are within the manuscript and its Supporting Information files.

## Abstract

Soil-transmitted helminths (STH) infect 1.5 billion people and countless animals worldwide. In Australian Indigenous communities, STH infections have largely remained endemic despite control efforts, suggesting reservoirs of infection may exist. Dogs fulfil various important cultural, social and occupational roles in Australian Indigenous communities and are populous in these settings. Dogs may also harbour zoonotic STHs capable of producing morbidity and mortality in dogs and humans. This review provides an overview of human and zoonotic STH infections, identifies the Australian Indigenous locations affected and the parasite species and hosts involved. The meta-analysis provides estimates of individual study and pooled true prevalence of STH infections in Australian Indigenous communities and identifies knowledge gaps for further research on zoonotic or anthroponotic potential. A systematic literature search identified 45 eligible studies documenting the presence of *Strongyloides stercoralis*, *Trichuris trichiura*, *Ancylostoma caninum*, *Ancylostoma duodenale*, *Ancylostoma ceylanicum*, undifferentiated hookworm, and *Ascaris lumbricoides*. Of these studies, 26 were also eligible for inclusion in meta-analysis to establish true prevalence in the light of imperfect diagnostic test sensitivity and specificity by Rogan-Gladen and Bayesian methods. These studies revealed pooled true prevalence estimates of 18.9% (95% CI 15.8–22.1) for human and canine *S. stercoralis* infections and 77.3% (95% CI 63.7–91.0) for canine *A. caninum* infections indicating continued endemicity, but considerably more heterogenous pooled estimates for canine *A. ceylanicum* infections, and *A. duodenale*, undifferentiated hookworm and *T. trichiura* in humans. This review suggests that the prevalence of STHs in Australian Indigenous communities has likely been underestimated, principally based on imperfect diagnostic tests. Potential misclassification of hookworm species in humans and dogs due to outdated methodology, also obscures this picture. High-quality contemporary studies are required to establish current true prevalence of parasite species in all relevant hosts to guide future policy development and control decisions under a culturally sound One Health framework.

**Funding:** CR and PZ are recipients of a University of Melbourne Research and Training Program Scholarship. CR is a Lowitja Institute Postgraduate Scholarship recipient. The funders had no role in study design, data collection and analysis, decision to publish, or preparation of the manuscript.

**Competing interests:** The authors have declared that no competing interests exist.

## Author summary

Soil-transmitted helminths include hookworms, threadworms, whipworms and round-worms. These worms may infect different hosts including humans and dogs, and some species are zoonotic, meaning that they are able to transmit between animals and humans. In many Australian Indigenous communities, people remain infected with these worms at high rates compared to other parts of the country despite various control strategies. Resource and health literacy inequalities are primary drivers for these differences. However, the potential for dogs to act as reservoirs for zoonotic worm infections in humans must also be considered. For this reason, it's important to create a clear picture of the level of infection by location and host. Given that tests used to establish prevalence can produce false positive or negative results, we performed a meta-analysis allowing comparison of true prevalence estimates by location and host, regardless of the test used. This review suggests that threadworm and dog hookworm remain endemic in Australian Indigenous communities, though a gap exists to accurately inform the prevalence of the other worms. It also highlights the need for One Health strategies in research, policy and control where humans, all animal hosts and the environment are considered in a culturally relevant way.

## Introduction

Soil-transmitted helminths (STHs) are Neglected Tropical Diseases (NTDs) and infect an estimated 1.5 billion people and countless animals worldwide [1]. In Australian Indigenous communities, it is necessary to develop an understanding of not only the STHs of concern to humans, but also other hosts such as dogs, that may act as zoonotic reservoirs for STHs that can mature to adulthood in humans, as well as environmental factors which may contribute to transmission. The importance of the relationship between people and canines is clear in the great significance and diverse roles that dogs hold in these communities; as companions, hunting partners, spiritual guardians and members of the intricate kinship system [2–4].

STHs relevant to humans in Australia include hookworms (including *Ancylostoma duodenale*, *Ancylostoma ceylanicum* and *Necator americanus*), threadworms (*Strongyloides stercoralis*), whipworms (*Trichuris trichiura*) and *Ascaris lumbricoides*. These genera are endemic throughout nearby Oceanic countries with hookworms, in particular, constituting overall prevalence of 48% among the 9.6 million humans (excluding Australia and New Zealand) residing in this region [5,6]. Papua New Guinea, which has a climate similar to that of Northern Australia, is overrepresented in terms of hookworm and appears to drive this regional prevalence, with an estimated prevalence of 60.6% based on a Global Burden of Disease study by Pullan and colleagues in 2010 [5].

Several zoonotic STHs of canines and felines are able to mature in human hosts including the hookworms *A. ceylanicum* and *A. caninum* [6–9]. *A. ceylanicum* has been established as the second most common hookworm infecting humans in the Asia Pacific region, and it is the only zoonotic hookworm known to cause patent, egg-shedding infections in humans [8–11]. *A. caninum* has mostly been thought to be a parasite of dogs but can mature in humans forming non-patent infections [12–14]. Recent evidence of egg-shedding infections in humans however, suggest that patent infections are possible [15]. Strongyloidiasis is also a potential zoonosis, with *S. stercoralis* comprising two distinct genetic clades, one restricted to dogs and another infecting humans, non-human primates, dogs and cats [16,17]. Routine anthelmintic use in canine or human hosts along with sanitation may reduce environmental contamination

with larvae, reducing infection pressure on humans or canines sharing the same environment [17–20].

## Parasite biology

Several lifecycle and transmission features are common to STH species found in Australia, along with some important differences, with both helping to inform diagnosis, treatment and control strategies as part of a One Health approach for these important STH species. The most important commonality of all STHs is that infectivity to the next host is reliant on an essential period of development in the environment (usually soil) ranging from less than one to over 14 weeks, depending on environmental conditions such as temperature and humidity [21,22]. Third stage filariform larvae of all hookworm species and *Strongyloides* can infect their host by percutaneous penetration of exposed skin in contact with contaminated matter. People who regularly walk barefoot outdoors, as regularly observed in remote Australian Indigenous communities, demonstrate higher risk of hookworm and *Strongyloides* infection [23]. A Malaysian study found that humans who routinely walked barefoot outdoors were 5.6 (95% CI 2.9–10.7) times more likely to be infected with hookworms compared with those who routinely wore shoes [24]. *T. trichiura* and *A. lumbricoides* infect their human definitive host by ingestion of substrates contaminated with embryonated eggs. Third-stage filariform larvae of some *Ancylostoma* spp. can also infect their hosts via ingestion of contaminated matter. Dogs may also become infected with *A. caninum* by the ingestion of paratenic hosts such as rats or mice [25].

Once infected, lifecycles and associated symptoms differ between STH genera. Adults of hookworms and *Strongyloides* spp. reside in the small intestines, and *Trichuris* spp. in the colon of dogs and humans, thus producing signs that range from subclinical to severe and include intestinal haemorrhage, anaemia, abdominal discomfort and diarrhoea [6,26–29]. Human infections with the zoonotic hookworm *A. caninum* have, in some cases, presented clinically with eosinophilia and acute eosinophilic enterocolitis, which has thus far been found to be the result of only a single pre-adult worm in all closely examined cases [13,14]. While this suggests that patent infections are not possible in humans, past, as well as more recent evidence demonstrating egg-shedding, challenge this paradigm [15,30]. Hookworms and *Strongyloides* spp. can also produce dermatological symptoms that include raised itchy rashes, urticaria and pulmonary symptoms of coughing as a result of percutaneous and hepatopulmonary migration [31]. *Ascaris* may also stimulate coughing related to its hepatopulmonary migration following ingestion, but otherwise usually only causes mild abdominal discomfort in light burdens [32]. The autoinfective cycle of *S. stercoralis* and its ability for massive synchronous larval emergence (hyperinfection) in immunocompromised or immunosuppressed patients can lead to mortality rates as high as 87% in both humans and dogs [20,33–36]. This is important because while most *S. stercoralis* cases are self-limiting in dogs, the non-specific clinical presentation that may manifest in both dogs and humans, can potentially progress rapidly to a state of hyperinfection or disseminated strongyloidiasis, which constitute a poor prognosis [33,34,37].

## Environmental factors

Several factors may favour environmental contamination with eggs and larvae thereby facilitating infection in remote Indigenous community settings. The hot and humid climate of the tropical or subtropical North of Australia favour development and survival of infective stages of STHs in the soil, with dogs from tropical climates found to be 5.6 (95% CI 3.3–9.5) times more likely to be infected with hookworms compared to dogs from non-tropical climate zones in Australia [38]. Another study found a greater proportion of Australian dog parks in tropical

regions (91.7%) to be contaminated with STHs compared with those in temperate (39.5%) and subtropical regions (33.3%) [30,39]. The same study found 44.5%, 4.8% and 0.9% of dog parks in tropical, subtropical and temperate regions to be contaminated with *A. caninum*, respectively [39]. In the final report of the Australian Hookworm Campaign 1919–1924, Sweet described an increase in detected hookworm prevalence north of 32 degrees south latitude and in areas with greater than 1016 mm precipitation annually [40].

## Host and socio-economic factors

Large populations of owned and stray dog and cat populations in Indigenous communities contribute to large burdens of environmental contamination with STH eggs or larvae. Close and frequent contact between hosts in shared environments increases risk of zoonotic exchange in these settings, as shown in other countries such as Malaysia in which close contact with dogs or cats resulted in a 2.9 times increase in the risk of hookworm infection [24]. These risks and the importance of the human-animal bond in Australian Indigenous communities necessitates a culturally relevant One Health understanding of these infections [30,41].

Indigenous Australians suffer from disproportionately high rates of NTDs compared to their non-Indigenous counterparts [42,43]. Historically, around the time of the Australian Hookworm Campaign in the 1920s, Indigenous populations were found to have higher prevalence of both hookworms (62% vs 15%) and *Strongyloides* (0.8% vs 0.03%) compared with their non-Indigenous counterparts [6,27]. More recently, efforts towards control and prevention of these parasites in Indigenous communities have been largely ineffective, inconsistent or completely lacking [6,44]. Resource and health-literacy inequalities faced in these remote communities along with overcrowding also contribute to greater risk in several disease categories including STH infections [45]. At a time where government efforts to close the health inequality gap between Indigenous and non-Indigenous Australians is not on track to be achieved by the target of 2031, understanding sources of infection and risks of transmission are particularly important [46].

In order to understand the risk of transmission and identify targets for prevention and control, it is important first to define a clear, One-Health-focussed picture of parasite distribution in human, canine and feline host species which share common environments. With this background, the aim of this manuscript is to systematically review the literature and provide an overview of research conducted in this area, understand the locations, parasite and host species involved, provide an estimate of the individual study and pooled true prevalence of STH infections in Australian Indigenous communities and to identify knowledge gaps which may shed further light on the potential for zoonotic or anthroponotic spread. In this way a more holistic, One Health perspective may be attained rather than focussing control efforts at any one location, parasite or host in isolation. Such information has the potential to guide effective and culturally relevant policy and practices, drive community involvement and deliver preventive and curative solutions for problems plaguing Australian Indigenous communities for decades.

## Methods

### Search protocol

The literature search was focussed on soil-transmitted helminth infections (outcome) in humans and zoonotic STH infections in canines and felines in Australian Indigenous communities (population). In March 2020, veterinary, medical and public health databases were searched including PubMed, Embase, Directory of Open Access Journals, Web of Science, CAB Abstracts, Scopus, Medline, Biosis, APAIS Health, CINAHL, EBM Reviews and Google Scholar. Search terms included a combination of parasite species or disease manifestations (i.e.

**Table 1. Systematic review search terms [a].**

| Domain | Search terms |
|---|---|
| Parasite/ disease | *Ancylostoma*, "*Ancylostoma duodenale*", "*Ancylostoma ceylanicum*", "*Ancylostoma caninum*", ancylostomiasis, *Ascaris*, "*Ascaris lumbricoides*", ascariasis, helminthiasis, Helminth, Hookworm, "Hookworm infection", "Intestinal helminth", *Necator*, "*Necator americanus*", Nematode, "Soil-transmitted helminthiasis", "Soil-transmitted helminth", *Strongyloides*, "*Strongyloides stercoralis*", strongyloidiasis, Threadworm, *Trichuris*, "*Trichuris trichiura*", trichuriasis, Worms, Parasites |
| Location | "Australian Aboriginal community", "Aboriginal community", "Indigenous Australian", "Indigenous community" |
| Population | Dog, Dingo, Canine, Canis, Cat, Feline, Felis, Human, Adult, Child, Children, Infant |

[a] search terms combined using Boolean logic, with row terms combined using 'or' statements and 'and' statements used to combine terms between rows

"*Ancylostoma*" OR "Ancylostomiasis" . . .), AND location (i.e. "Australian Aboriginal community" OR "Aboriginal community" . . .) AND study population (i.e. "Dog" OR "Human" . . .) as detailed in Table 1. Reference lists of eligible articles and other works of frequent authors were also searched, and further articles included if eligible. The protocol was registered with the international prospective register of systematic reviews (PROSPERO), in accordance with PRISMA guidelines. Due to PROSPERO database constraints, human (PROSPERO CRD42020166266) and animal (PROSPERO CRD42020165388) protocols were registered separately with mutual references in each.

## Exclusion criteria

Full search result lists and exclusion assessments were stored in a proprietary spreadsheet (Microsoft Excel v. 1908, Microsoft Corporation, Redlands, California) and study citation details and digital copies of published papers managed using Mendeley Desktop (v. 1.19.8, Elsevier). After removing duplicates, studies were excluded if the work was not conducted in Australian Aboriginal or Torres Strait Islander communities; these included review articles, editorials, commentaries, letters, conference proceedings, or abstracts that did not contain data on the presence of soil-transmitted helminth species *A. duodenale*, *A. caninum*, *A. ceylanicum*, *A. lumbricoides*, *Necator americanus*, *S. stercoralis* or *T. trichiura*; study data were not from canine, feline or human hosts; data from another study was repeated or if a diagnostic methodology was not reported. When two or more studies containing repeated data were compared, those with a smaller subset were excluded. Studies using a case-control methodology were only included if STH infection was not a defined inclusion criterion for cases.

Studies were excluded from quantitative meta-analysis if they contained pooled results from Indigenous community and non-Indigenous community locations that could not be disaggregated; they did not report a quantitative measure of disease frequency such as prevalence or incidence or did not provide sufficient data to allow prevalence or incidence to be calculated; or if they did not provide sufficient detail of diagnostic methodology to allow sensitivity and specificity data to be linked. If uncertainties arose surrounding the inclusion or exclusion of a study, this was discussed amongst the research team and a consensus was formed.

## Data extraction

The following data were extracted from each eligible study (S1 Table): title, authors, publication year, study objectives, study site(s), study periods, study design, host species, sample type, sample size, parasite species, diagnostic technique, number of positive samples, factors affecting prevalence or incidence and the presence of an ethics statement. In cases of intervention

studies, pre-treatment prevalence or incidence was included in order to be more comparable to observational studies.

The presence of bias in the reported data was assessed for all studies following the method described by Hoy [47]. In brief, this method involves assessing studies across the domains of selection, nonresponse, measurement and analysis leading to an overall summary estimate of study bias. This method was selected for its relevance to studies of prevalence and ease of use, and was modified to consider Indigenous human, dog or cat populations rather than the national population. Ten percent of studies were extracted and critically appraised in duplicate by two research team members including the primary author to check for agreement. The remainder of the studies were then extracted and assessed by the primary author and incorporated into S1 Table.

## Data analyses

Prevalence data were extracted from the eligible studies and expressed as apparent prevalence (AP), i.e., the reported number of test-positive individuals divided by the total number of individuals tested. Due to the use of several different diagnostic techniques, each of them with imperfect sensitivity and specificity, variations in AP may arise by means of differences in diagnostic test accuracy. The AP of each study was therefore reanalysed as true prevalence (TP) taking into account imperfect test diagnostic sensitivities and specificities in order to allow comparisons of prevalence across studies. Sensitivity and specificity data for diagnostic tests used in the included studies were extracted from peer-reviewed articles which calculated these measures. Where multiple estimates of sensitivity and specificity for a given diagnostic test were found in the literature, an average of these measures was taken. Where no sensitivity and specificity data could be found in peer-reviewed literature, expert opinion was sought from two veterinary parasitologists, with averages taken from these opinions. *S. stercoralis* serologic tests can vary widely in sensitivity and specificity based on cut-offs, study populations, reference methods and test methods [48]. Five of the ten studies included for meta-analysis stated the use of IgG ELISA [49–52]. A further four studies [53–56] published by common authors did not refer to serological methods used but utilised the same study population and adjacent study timeframes, and thus are assumed to have used the same IgG ELISA assay. Another study [23] did not specifically state the serological method used to diagnose *S. stercoralis* exposure but referenced a study in which indirect immunofluorescence test (IFAT) was utilised. Given that these methods have very similar reported sensitivities and specificities in the source studies referenced in Table 2, the average diagnostic parameters were used in calculations of TP.

Sensitivity and specificity estimates and their sources are detailed in Table 2.

TP estimates were calculated using the method described by Rogan and Gladen [72] implemented in the contributed epiR package (v 2.0.39, Stevenson *et al.*, 2021) in R version 4.1.2 (R Core Team, 2021). In studies where the AP was less than (1—diagnostic test specificity) the Rogan Gladen estimate of TP was less than zero [73]. Similarly, if AP was greater than the diagnostic test sensitivity the Rogan Gladen estimate of TP was greater than unity. For these studies a Bayesian approach was used to estimate TP as described by Messam et al. [74] Here, estimates of the distributional form of diagnostic sensitivity, specificity and TP were used as priors and a Markov chain Monte Carlo approach used to combine these prior estimates with the empirical data to return a posterior estimate of TP and its 95% credible interval [75]. Distributions for sensitivity and specificity were taken from confidence intervals from the sources listed in Table 2 or were estimated within a 95% confidence interval by expert opinion if no published data were available. Estimates of the 'true' number of positive samples as a proportion of

**Table 2. Sensitivity and specificity data for tests used in studies included for meta-analysis.**

| Parasite | Diagnostic test | Sensitivity (%) | Specificity (%) | Source |
|---|---|---|---|---|
| Hookworm | Conventional PCR (cPCR) | 84.7 | 87.6 | [57,58] |
|  | Direct faecal smear (DS) | 16.3 | 100[a] | [59] |
|  | Formol-ether concentration (FE) | 49.9 | 100 | [60,61] |
|  | Morphology of adult worms (MpAd) | 13.8 | 100 | [62] |
|  | Saturated salt flotation (SSF) | 37.9 | 100 [a] | [39] |
| *Trichuris trichiura* | Direct faecal smear (DS) | 14.9 | 100 [a] | [59] |
|  | Formol-ether concentration (FE) | 63 | 100 | [60,61] |
|  | Saturated salt flotation (SSF) | 63.9 | 96.4 | [63] |
| *Strongyloides* spp. | Direct faecal smear (DS) | 30 | 100 [a] | [64–66] |
|  | ELISA serology (SE) | 82.4 | 92.5 | [67–69] |
|  | Formol-ether concentration (FE) | 34 | 100 | [66] |
|  | qPCR | 88.9 | 92.7 | [70] |
|  | Zinc sulphate flotation (SSF) | 9.3 | 100 [a] | [71] |

[a] Assumed specificity when no data available from peer-reviewed sources. Based on very low likelihood of false positives occurring via these diagnostic techniques

the total number of samples tested for each study (calculated using the Rogan-Gladen or Bayesian approach described above) allowed us to calculate a summary estimate of the true prevalence of soil-transmitted helminths across all of the studies included in the systematic review using the contributed metafor (v 3.0–2, Viechtbauer, 2021) package in R. True prevalence estimates and their 95% confidence intervals were plotted as a forest plot using the contributed forestplot (v 2.0.1, Gordon and Lumley, 2021) package in R.

The easting and northing location of the centroid of the area covered by each of the studies included in this review were plotted as a symbol map with colour and shape used to indicate host species. Geospatial analyses and mapping were carried out using the contributed ggplot2 (v 3.3.5, Wickham *et al.*, 2021) package in R. Mapping shapefile data were obtained from GADM database of Global Administrative Areas (v 2.8, GADM, 2018). Location details for Indigenous communities were obtained from the Australian Government National Indigenous Australians Agency [76].

## Results

The protocol for identifying, screening and excluding studies along with the numbers of studies excluded based on the criteria detailed in the methods section is shown in Fig 1. There were 327 studies initially identified after duplicates were removed. Following initial screening and full-text assessment for eligibility, 45 studies were eligible for qualitative synthesis. Of those, 26 fitted the criteria for inclusion in the meta-analysis.

The distribution of included studies by decade of publication is shown in Fig 2. Studies included in both quantitative and qualitative synthesis are included in this figure. While studies were identified for inclusion from 1921 to 2020, no studies were identified in the period from 1925 to 1969. Most included studies were published in the 1990s (31%) and 2010s (42%) with only 11% of studies published in the intervening decade.

The geospatial distribution of STH infection data from the included studies is shown in Fig 3. These studies reported the presence of STH infections in Indigenous communities in the Northern Territory (21 of 45 studies; 47%), Western Australia (15 of 45 studies; 33%), Queensland (14 of 45 studies; 31%) and New South Wales (4 of 45 studies; 9%). While some studies

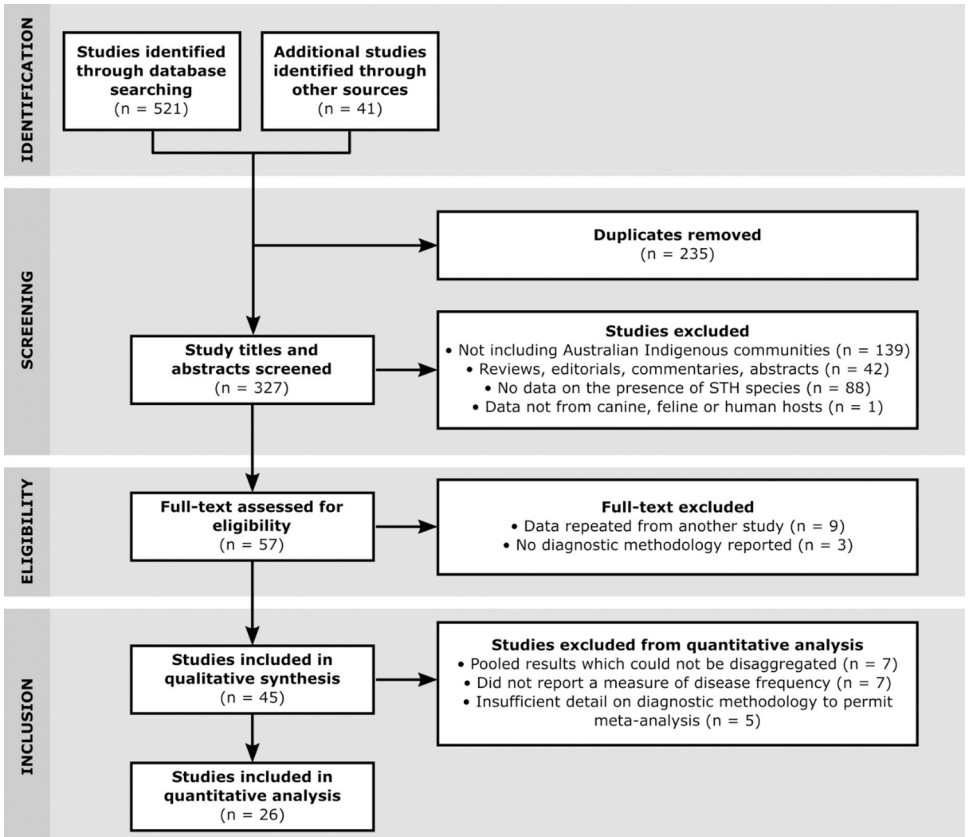

**Fig 1. PRISMA flow diagram of the search, screening for eligibility and inclusion in qualitative and quantitative synthesis.**

reported data on locations in multiple states or territories, no studies meeting inclusion criteria reported data on STH infections in South Australia, Victoria, Tasmania or the Australian Capital Territory. Some of the most studied locations of the included studies include the Kimberley (12 of 45 studies; 27%), Far North Queensland (7 of 45 studies; 16%), Arnhem Land (7 of 45 studies; 16%), and Alice Springs and surrounds (6 of 45 studies; 13%).

Where hookworms were identified by the presence of ova only, these studies were designated as 'hookworm', regardless of whether species were designated in the study as this differentiation is not possible by microscopy. *Ascaris lumbricoides* has been omitted from mapping as only one case was reported in a 1992 study from the Kimberley region in North-West Western Australia [77].

Indigenous communities were not exclusive in all studies, with seven studies reporting data from Indigenous and non-Indigenous community locations which could not be disaggregated. Eleven studies did not designate a specific Indigenous community location, either for the purpose of community privacy and anonymity or because state-wide or nationwide Indigenous community data were aggregated. Five studies featured sampling from either Alice Springs or Darwin hospitals, both of which service large areas and would include data from many Indigenous communities within a large radius of each hospital.

Most included studies were surveys (25 of 45; 56%), with the remainder of studies being case-control studies (9 of 45; 20%), cohort studies (7 of 45; 16%) and case reports (4 of 45; 9%). Sample sizes in the included studies had a median of 91.5 samples and a range of 2 to 9956.

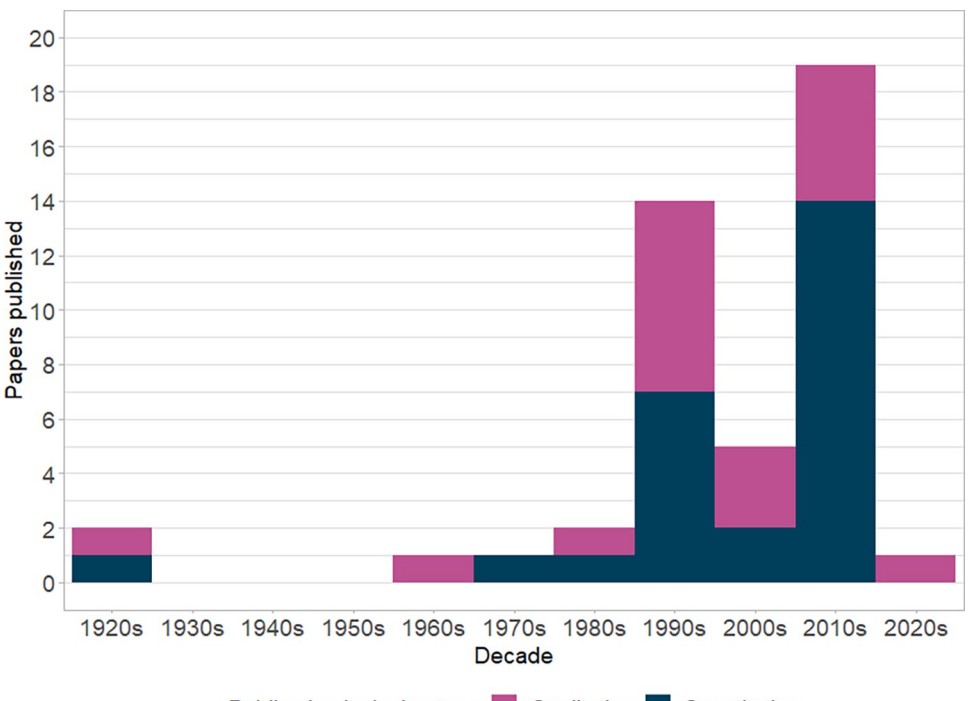

**Fig 2. Studies included in quantitative and qualitative synthesis published per decade.**

Only 52% of studies specified that ethical approval was sought, none of which were published prior to 1997. All included studies since 2010 reported human or animal ethics approval.

Human samples were the most common host sample type analysed in the included studies (37 of 45; 82%), followed by domestic dogs (11 of 45; 24%) and wild dogs/dingoes (3 of 45; 7%). For the purposes of this review the term 'domestic dog' refers to dogs living in an Indigenous community who may have been owned by an individual, a family, or a community more generally. Wild dogs and dingoes may be dogs living wild existences, dog/dingo hybrids or pure dingoes, though distinctions were not made in the included studies. While felines were included in the search protocol, no studies were identified which fitted the inclusion criteria.

Faecal samples were the most commonly featured sample type collected from hosts in the included studies (32 of 45; 71%), followed by blood (11 of 45; 24%), dog or wild dog/dingo intestinal content from necropsy specimens (5 of 45; 11%), surgical biopsies of sections of intestine in humans (3 of 45; 7%), soil from in and around Indigenous communities (1 of 45; 2%), and sputum (1 of 45; 2%).

A wide range of diagnostic tests were used to detect STH infections with serology (SE) most commonly used, though only in *S. stercoralis* studies (12 of 45; 27%), followed by direct smear or wet mount microscopy (DS) (9 of 45; 20%), saturated salt flotations (SSF) (8 of 45; 18%), conventional PCR (cPCR) (5 of 45; 11%), formol-ether concentration method microscopy (FE) (3 of 45; 7%), morphological identification of adult worms from necropsy specimens in dogs or surgical biopsies in humans (MpAd) (3 of 45; 7%), and quantitative PCR (qPCR) (1 of 45; 2%).

Data on the presence of *S. stercoralis* was reported in the greatest proportion of studies (28 of 45; 62%), followed by undifferentiated hookworm species (17 of 45; 38%), *T. trichiura* (10 of 45; 22%), *A. caninum* (8 of 45; 18%), *A. duodenale* (6 of 45; 13%), *A. ceylanicum* (3 of 45; 7%), and *A. lumbricoides* (1 of 45; 2%). No studies identified in this review featured data on the

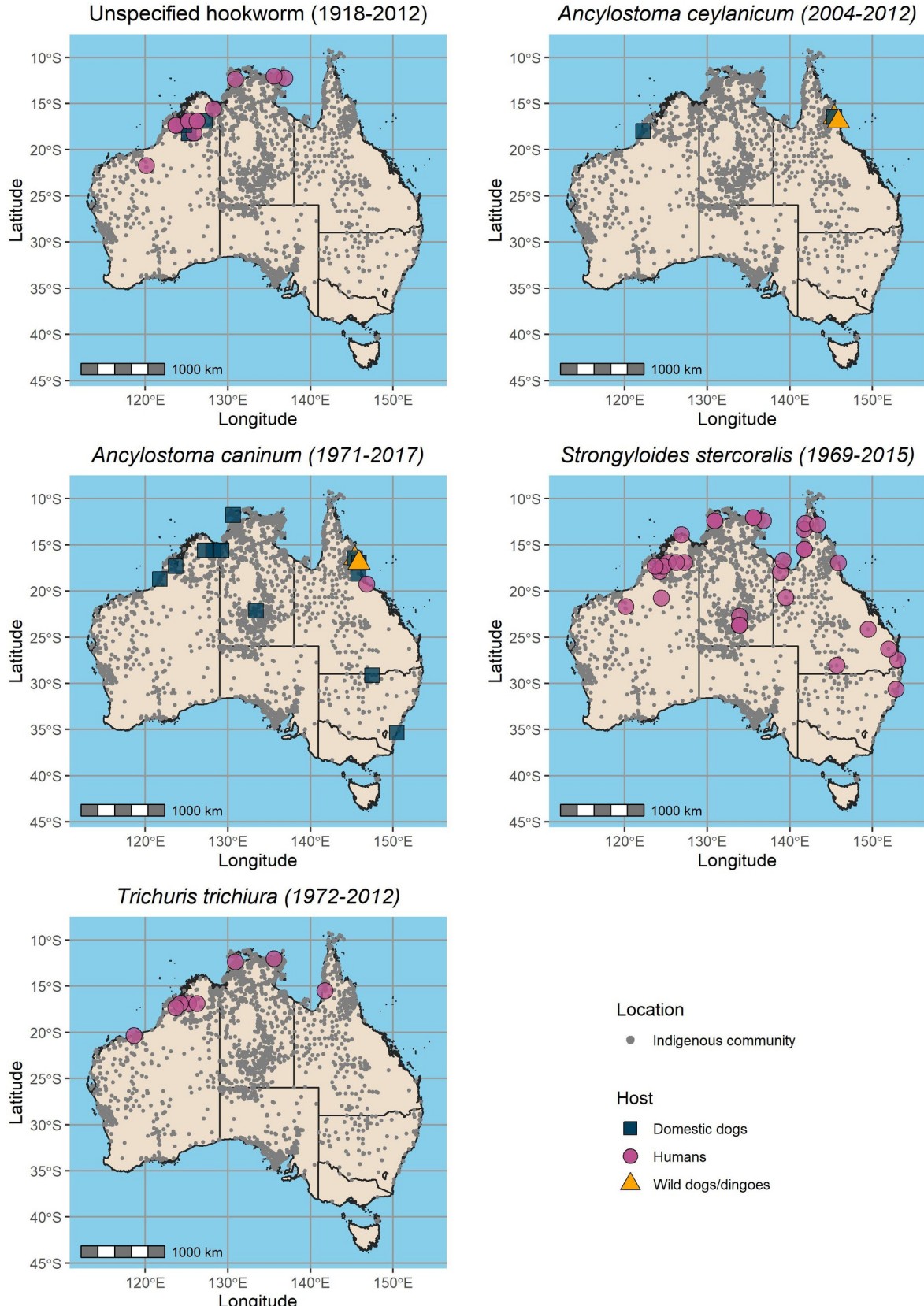

**Fig 3. Detected STH species and sampling date spans in domestic dog, wild dog, dingo and human hosts in relation to Australian Indigenous community locations.** Maps developed in R using basemap shapefiles from the Database of Global Administrative Areas (GADM). https://geodata.ucdavis.edu/gadm/gadm4.1/shp/gadm41_AUS_shp.zip.

presence of *N. americanus*. Fig 4 presents individual study TPs with confidence or credible intervals (where appropriate) for each of these parasites, with each line a distinct reported prevalence from a different host, diagnostic technique or Indigenous community location. Fifty-six out of 75 (75%) of TP measures were greater than their originally reported AP. Studies for each parasite are summarised into pooled TP estimates with 95% confidence intervals.

### Ancylostoma caninum

For some species as well as for unidentified hookworm infections, prevalence estimates varied markedly. In the case of *A. caninum*, however, the pooled TP estimate of 77% and confidence interval presented in a tighter band of 64% to 91%, reflecting most of the contributing individual estimates. Lower estimates such as those from the study by Slapeta *et al.* (2015) all came from studies that used saturated salt flotation diagnostic methods and were from more arid locations which may be less suited to environmental development of hookworm larvae (S1 Table). Almost all other individual study estimates were from tropical or subtropical regions such as the Kimberley or Far North Queensland. Studies not suitable for meta-analysis included a study where 48% of dog faecal samples from a Kimberley community were found to contain *A. caninum*, though these were reported only when *Giardia duodenalis*—the focus of the study—was absent in samples [78]. While TP data in humans is notably absent for *A. caninum* in Fig 4, two sets of case studies reported eosinophilic enteritis caused by adult worms in Northern Queensland [13,14].

### Ancylostoma ceylanicum

Two studies of *A. ceylanicum* by Smout et al. (2017, 2018) reported widely differing prevalence estimates despite these samples being collected from the same region in Far North Queensland. It should be noted that sampling in one of these studies featured collection of soil from around communities rather than directly from animals. These limited number of studies represent some of the smallest sample sizes of those included in these analyses, which contributes to both their wide individual confidence intervals and the pooled TP estimate of 26% (95% CI 2% to 50%). One study unsuitable for meta-analysis detected *A. ceylanicum* in an unreported number of samples in Broome [38].

### Ancylostoma duodenale

*A. duodenale* produced a pooled TP estimate of 52% and a wide confidence interval of 18% to 87%, with no individual TP estimates falling within this range. Individual estimates fell at the extremes of the TP scale. Two of these studies, both of which make up prevalence estimates at the lower extreme of the scale, have a higher risk of bias due to poor representation of the population in the sample group. The study by Meloni *et al.* (1993) sampled children only and the study by Jones did not use random sampling or include details on the age of participants [79,80]. One study not suitable for meta-analysis from East Arnhem Land in the Northern Territory found an AP of 15%, though risk of bias was also high due to poor sample selection [81]. All other studies were from the Kimberley region in Western Australia. It should also be noted that these studies are significantly dated in comparison to other included studies, with the most recent being published in 1998.

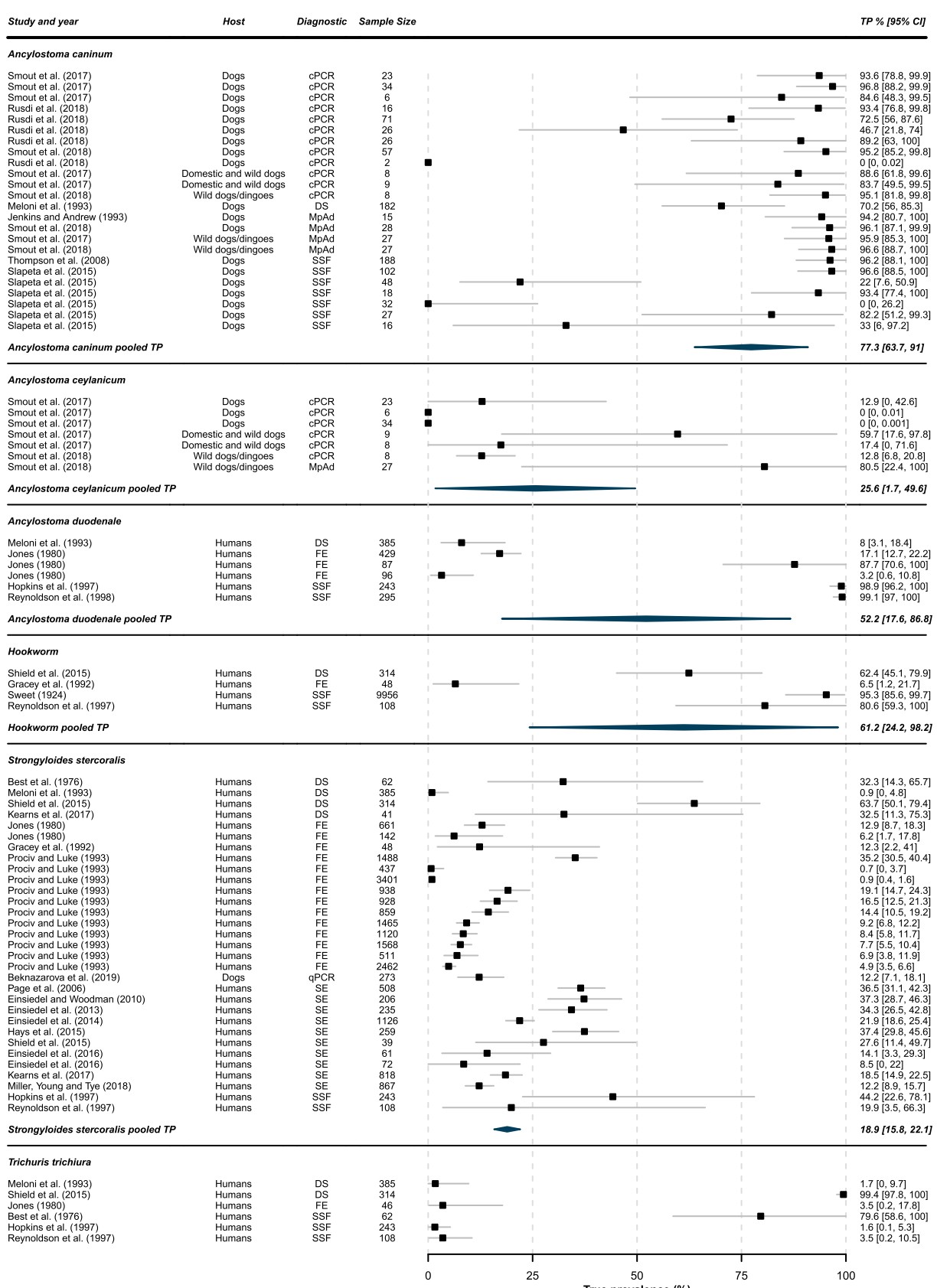

| Study and year | Host | Diagnostic | Sample Size | | TP % [95% CI] |
|---|---|---|---|---|---|
| **Ancylostoma caninum** | | | | | |
| Smout et al. (2017) | Dogs | cPCR | 23 | | 93.6 [78.8, 99.9] |
| Smout et al. (2017) | Dogs | cPCR | 34 | | 96.8 [88.2, 99.9] |
| Smout et al. (2017) | Dogs | cPCR | 6 | | 84.6 [48.3, 99.5] |
| Rusdi et al. (2018) | Dogs | cPCR | 16 | | 93.4 [76.8, 99.8] |
| Rusdi et al. (2018) | Dogs | cPCR | 71 | | 72.5 [56, 87.6] |
| Rusdi et al. (2018) | Dogs | cPCR | 26 | | 46.7 [21.8, 74] |
| Rusdi et al. (2018) | Dogs | cPCR | 26 | | 89.2 [63, 100] |
| Smout et al. (2018) | Dogs | cPCR | 57 | | 95.2 [85.2, 99.8] |
| Rusdi et al. (2018) | Dogs | cPCR | 2 | | 0 [0, 0.02] |
| Smout et al. (2017) | Domestic and wild dogs | cPCR | 8 | | 88.6 [61.8, 99.6] |
| Smout et al. (2017) | Domestic and wild dogs | cPCR | 9 | | 83.7 [49.5, 99.5] |
| Smout et al. (2018) | Wild dogs/dingoes | cPCR | 8 | | 95.1 [81.8, 99.8] |
| Meloni et al. (1993) | Dogs | DS | 182 | | 70.2 [56, 85.3] |
| Jenkins and Andrew (1993) | Dogs | MpAd | 15 | | 94.2 [80.7, 100] |
| Smout et al. (2018) | Dogs | MpAd | 28 | | 96.1 [87.1, 99.9] |
| Smout et al. (2017) | Wild dogs/dingoes | MpAd | 27 | | 95.9 [85.3, 100] |
| Smout et al. (2018) | Wild dogs/dingoes | MpAd | 27 | | 96.6 [88.7, 100] |
| Thompson et al. (2008) | Dogs | SSF | 188 | | 96.2 [88.1, 100] |
| Slapeta et al. (2015) | Dogs | SSF | 102 | | 96.6 [88.5, 100] |
| Slapeta et al. (2015) | Dogs | SSF | 48 | | 22 [7.6, 50.9] |
| Slapeta et al. (2015) | Dogs | SSF | 18 | | 93.4 [77.4, 100] |
| Slapeta et al. (2015) | Dogs | SSF | 32 | | 0 [0, 26.2] |
| Slapeta et al. (2015) | Dogs | SSF | 27 | | 82.2 [51.2, 99.3] |
| Slapeta et al. (2015) | Dogs | SSF | 16 | | 33 [6, 97.2] |
| **Ancylostoma caninum pooled TP** | | | | | **77.3 [63.7, 91]** |
| **Ancylostoma ceylanicum** | | | | | |
| Smout et al. (2017) | Dogs | cPCR | 23 | | 12.9 [0, 42.6] |
| Smout et al. (2017) | Dogs | cPCR | 6 | | 0 [0, 0.01] |
| Smout et al. (2017) | Dogs | cPCR | 34 | | 0 [0, 0.001] |
| Smout et al. (2017) | Domestic and wild dogs | cPCR | 9 | | 59.7 [17.6, 97.8] |
| Smout et al. (2017) | Domestic and wild dogs | cPCR | 8 | | 17.4 [0, 71.6] |
| Smout et al. (2018) | Wild dogs/dingoes | cPCR | 8 | | 12.8 [6.8, 20.8] |
| Smout et al. (2018) | Wild dogs/dingoes | MpAd | 27 | | 80.5 [22.4, 100] |
| **Ancylostoma ceylanicum pooled TP** | | | | | **25.6 [1.7, 49.6]** |
| **Ancylostoma duodenale** | | | | | |
| Meloni et al. (1993) | Humans | DS | 385 | | 8 [3.1, 18.4] |
| Jones (1980) | Humans | FE | 429 | | 17.1 [12.7, 22.2] |
| Jones (1980) | Humans | FE | 87 | | 87.7 [70.6, 100] |
| Jones (1980) | Humans | FE | 96 | | 3.2 [0.6, 10.8] |
| Hopkins et al. (1997) | Humans | SSF | 243 | | 98.9 [96.2, 100] |
| Reynoldson et al. (1998) | Humans | SSF | 295 | | 99.1 [97, 100] |
| **Ancylostoma duodenale pooled TP** | | | | | **52.2 [17.6, 86.8]** |
| **Hookworm** | | | | | |
| Shield et al. (2015) | Humans | DS | 314 | | 62.4 [45.1, 79.9] |
| Gracey et al. (1992) | Humans | FE | 48 | | 6.5 [1.2, 21.7] |
| Sweet (1924) | Humans | SSF | 9956 | | 95.3 [85.6, 99.7] |
| Reynoldson et al. (1997) | Humans | SSF | 108 | | 80.6 [59.3, 100] |
| **Hookworm pooled TP** | | | | | **61.2 [24.2, 98.2]** |
| **Strongyloides stercoralis** | | | | | |
| Best et al. (1976) | Humans | DS | 62 | | 32.3 [14.3, 65.7] |
| Meloni et al. (1993) | Humans | DS | 385 | | 0.9 [0, 4.8] |
| Shield et al. (2015) | Humans | DS | 314 | | 63.7 [50.1, 79.4] |
| Kearns et al. (2017) | Humans | DS | 41 | | 32.5 [11.3, 75.3] |
| Jones (1980) | Humans | FE | 661 | | 12.9 [8.7, 18.3] |
| Jones (1980) | Humans | FE | 142 | | 6.2 [1.7, 17.8] |
| Gracey et al. (1992) | Humans | FE | 48 | | 12.3 [2.2, 41] |
| Prociv and Luke (1993) | Humans | FE | 1488 | | 35.2 [30.5, 40.4] |
| Prociv and Luke (1993) | Humans | FE | 437 | | 0.7 [0, 3.7] |
| Prociv and Luke (1993) | Humans | FE | 3401 | | 0.9 [0.4, 1.6] |
| Prociv and Luke (1993) | Humans | FE | 938 | | 19.1 [14.7, 24.3] |
| Prociv and Luke (1993) | Humans | FE | 928 | | 16.5 [12.5, 21.3] |
| Prociv and Luke (1993) | Humans | FE | 859 | | 14.4 [10.5, 19.2] |
| Prociv and Luke (1993) | Humans | FE | 1465 | | 9.2 [6.8, 12.2] |
| Prociv and Luke (1993) | Humans | FE | 1120 | | 8.4 [5.8, 11.7] |
| Prociv and Luke (1993) | Humans | FE | 1568 | | 7.7 [5.5, 10.4] |
| Prociv and Luke (1993) | Humans | FE | 511 | | 6.9 [3.8, 11.9] |
| Prociv and Luke (1993) | Humans | FE | 2462 | | 4.9 [3.5, 6.6] |
| Beknazarova et al. (2019) | Dogs | qPCR | 273 | | 12.2 [7.1, 18.1] |
| Page et al. (2006) | Humans | SE | 508 | | 36.5 [31.1, 42.3] |
| Einsiedel and Woodman (2010) | Humans | SE | 206 | | 37.3 [28.7, 46.3] |
| Einsiedel et al. (2013) | Humans | SE | 235 | | 34.3 [26.5, 42.8] |
| Einsiedel et al. (2014) | Humans | SE | 1126 | | 21.9 [18.6, 25.4] |
| Hays et al. (2015) | Humans | SE | 259 | | 37.4 [29.8, 45.6] |
| Shield et al. (2015) | Humans | SE | 39 | | 27.6 [11.4, 49.7] |
| Einsiedel et al. (2016) | Humans | SE | 61 | | 14.1 [3.3, 29.3] |
| Einsiedel et al. (2016) | Humans | SE | 72 | | 8.5 [0, 22] |
| Kearns et al. (2017) | Humans | SE | 818 | | 18.5 [14.9, 22.5] |
| Miller, Young and Tye (2018) | Humans | SE | 867 | | 12.2 [8.9, 15.7] |
| Hopkins et al. (1997) | Humans | SSF | 243 | | 44.2 [22.6, 78.1] |
| Reynoldson et al. (1997) | Humans | SSF | 108 | | 19.9 [3.5, 66.3] |
| **Strongyloides stercoralis pooled TP** | | | | | **18.9 [15.8, 22.1]** |
| **Trichuris trichiura** | | | | | |
| Meloni et al. (1993) | Humans | DS | 385 | | 1.7 [0, 9.7] |
| Shield et al. (2015) | Humans | DS | 314 | | 99.4 [97.8, 100] |
| Jones (1980) | Humans | FE | 46 | | 3.5 [0.2, 17.8] |
| Best et al. (1976) | Humans | SSF | 62 | | 79.6 [58.6, 100] |
| Hopkins et al. (1997) | Humans | SSF | 243 | | 1.6 [0.1, 5.3] |
| Reynoldson et al. (1997) | Humans | SSF | 108 | | 3.5 [0.2, 10.5] |

True prevalence (%)

Fig 4. **Forest plots for parasite species with pooled TP.** Individual true prevalence (TP) data with 95% confidence or credible intervals for each reported apparent prevalence within papers with pooled TP and 95% confidence intervals for each parasite species or designated as hookworm where species were not reported. For details on diagnostic tests and test accuracy see Table 2.

### Hookworm

Several of the aforementioned hookworm species may have contributed to the undifferentiated individual prevalence estimates seen from these studies, producing a pooled TP estimate of 61% (95% CI 24% to 98%). These studies also showed varying degrees of risk of bias. The study by Gracey *et al.* (1992), which had a notably lower TP estimate than other studies at 6%, featured sampling from children up to two years of age while the other studies feature a more representative sample group The report of the Australian Hookworm Campaign 1919–1924 by Sweet, while featuring a large sample size of 9,956 individuals, had a poor description of study methodology and no details on study participant demographic details or locations sampled in this nationwide survey. It is important to note also that the data contributing up the pooled TP estimate spans 91 years, during which time several efforts to combat hookworm infection have occurred and may have contributed to significant variation in prevalence.

Five other studies not suitable for meta-analysis featured data on the presence of undifferentiated hookworms, although only two of those could be disaggregated from non-Indigenous results and had denominators necessary to establish AP. A 1921 study by Lambert found an AP in humans of 77% across Queensland, although the description of sampling and methodology was poor and some data may overlap with that of the report by Sweet [40,82]. Another study by Fryar and Hagan found an AP of only 4%, though a poor description of methodology and sampling group leads to a moderate risk of bias [83].

### *Strongyloides stercoralis*

The larger number of included studies, many of which feature large sample sizes has permitted a pooled TP estimate of 19% with a relatively narrow confidence interval of 16% to 22% for *S. stercoralis* infection in Australian Indigenous communities. A study by Kearns *et al.* (2017) which investigated the effects of an ivermectin mass drug administration had a baseline TP estimate of 32% (95% CI 11% to 75%) when examining direct faecal smears from 41 children, but had a TP of 18% (95% CI 15% to 22%) when examining 818 serological samples from the wider community including adults. Similar differences are seen between the TP estimates derived by different diagnostic methods in the study by Shield *et al.* (2015). Studies by Best *et al.* (1976), Meloni *et al.* (1993), Gracey *et al.* (1992), and Prociv and Luke (1993) also have an increased risk of bias due to inclusion of children only [79,84–86,87]. Only one included study featured data on the presence of the parasite in dogs, although the aim of this study was focused on genetic characterisation of *Strongyloides* infections in dogs rather than prevalence [88].

Ten studies unsuitable for meta-analysis featured data on the presence of *S. stercoralis*. Of those, only three could be disaggregated from non-Indigenous results or had denominators necessary to determine AP. While these studies had AP estimates of 4%, 11% and 35% all had moderate to high risk of bias based on insufficient detail of study methodology and poor sampling strategy [77,83,89].

### *Trichuris trichiura*

Six studies were included for quantitative analysis of *T. trichiura* prevalence. These featured individual TP estimates at either extreme of the prevalence scale. The heterogeneity of

individual study data from locations in which *T. trichiura* infection was either highly prevalent or absent, is therefore likely to contribute to a misleading pooled TP estimate. Thus, a pooled

TP estimate was excluded from Fig 4. As mentioned above, the studies by Meloni *et al.* (1993) and Best *et al.* (1976) sampled only children and have a higher risk of bias. Four studies not suitable for meta-analysis included data on the presence of *T. trichiura*, though only two studies presented data where AP could be determined. Both these studies also featured data on *Strongyloides* and had moderate and high risk of bias for the reasons outlined above.

## Discussion

This review brings together for the first time, true prevalence data on STH infections capable of maturing in the gastrointestinal tract of dogs and humans in Australian Indigenous communities. The TP estimation methods used in this review allow comparisons to be made among studies using different diagnostic methodologies and permit the calculation of pooled TP estimates for each parasite species. While these pooled TP estimates and confidence intervals may be useful to extrapolate to other Indigenous communities across Australia as presented in Fig 3, there are several important caveats.

Firstly, it should be noted that while TP estimates have been calculated based on imperfect diagnostic tests, the sensitivity and specificity data used to calculate these are also imperfect themselves in that they are based on what was considered 'gold standard' at the time of calculation. In truth, even gold standard tests are imperfect and therefore TP calculations based on these should be viewed in this light.

The diagnostic methodologies in Fig 4 as well as S1 Table are expressed here as they were presented in the included papers. In the case of hookworm species *A. caninum* in dogs and *A. duodenale* in humans, coproscopic methods examining eggs including SSF, FE and DS are unable to determine species due to the identical morphological appearance of *Ancylostoma* spp. eggs [9,38]. This calls into question whether hookworm species were accurately identified by these methods or if the species were assumed based on which host the samples came from and should therefore be reclassified as undifferentiated hookworm infections. This would reclassify all papers with data on *A. duodenale* and more than half of those featuring *A. caninum*. Included studies featuring *A. ceylanicum*, by comparison, used methods such as cPCR and MpAd which permit species identification. While reclassification of these individual *A. caninum* and *A. duodenale* TP estimates may be more accurate, it is unlikely to clarify the pooled TP estimate of hookworms in Indigenous communities any further.

The importance of seasonal variation in the TP data collated in this study is unclear. Larval hypobiosis of *A. caninum* in dogs and humans and *A. duodenale* in humans has been described, with reactivation of egg shedding infections under favourable climatic conditions [6,90]. This phenomenon may have impacted prevalence in dry seasons, however timing of sample collection was rarely specified beyond specific years in the included studies making seasonal patterns difficult to establish.

Similarly, *S. stercoralis* may have intermittent shedding of larvae and uneven distribution of larvae in faeces, making faecal detection methods less sensitive [91,92]. Other challenges also arise regarding the method of detection of *S. stercoralis* infection. Studies included in this review featured both direct methods, in which the presence of larvae was detected in the faeces indicating current infection, and indirect methods, in which serology was conducted to detect antibodies against *Strongyloides*. While serology is useful for detecting response to treatment, the acute phase of infection may not elicit detectable antibody responses for several months, and sensitivity and specificity of serology varies with test cut-off values [48,51,66].

Hyperinfective states may also have unreliable results on serology, although these cases would likely be detectable with faecal samples [27,93]. While some of the included studies used agar plate culture as a follow-up to confirm *S. stercoralis* diagnosis, none used it as a primary means of diagnosis, nor did they use Baermann culture. While these are known to be more sensitive than most other faecal diagnostics, difficulties associated with sample storage and transport in order to keep larvae alive may have prevented the use of these methods [48].

*A. ceylanicum* and *T. trichiura* meta-analysis resulted in wide confidence intervals making meaningful conclusions on TP difficult. While *A. lumbricoides* was detected in one study, the single sample testing positive via undisclosed means and the high risk of bias in the study meant that this result was unlikely to be of significance [77].

While some studies are numerous and large enough to contribute to TP estimates with narrower confidence intervals, such as those of *A. caninum* and *S. stercoralis*, it is important that these prevalence data are acted upon. Alarmingly, while the TP data in Fig 4 cover time periods of 1993–2018 and 1976–2019 for *A. caninum* and *S. stercoralis*, respectively, neither demonstrate significant reductions in prevalence during those periods. The high TP of *A. caninum* in dogs presents a significant risk of environmental contamination in these remote communities where other factors such as tropical climate and barefoot walking also favour zoonotic infection or accidental ingestion. The vague and often mild symptoms of many of these STH infections has often led them to be omitted from differential diagnosis lists and undertreated in humans and animals alike [37,44,94,95]. Missed infections with hookworm or whipworm may lead to protracted anaemia, malaise and stunting in younger humans and animals, or in the case of *S. stercoralis*, hyperinfection can lead to severe disease and death. Routine testing and treatment programs which take a One Health approach considering all relevant hosts are required to establish more localised prevalence patterns both spatially and seasonally, and to catch infections early before serious sequelae occur.

Finally, few of the included studies included Indigenous community members as part of research teams. If meaningful policy development and community engagement is to be achieved, future studies must include Indigenous leadership and community involvement as part of surveillance and treatment efforts, as well as striving for the highest degrees of ethical standards.

## Conclusion

From the data presented in this review, it is likely that the prevalence of STH infections in Australian Indigenous communities has been underestimated in most cases based off imperfect diagnostic methodology. The use of coproscopic methods which cannot differentiate hookworm species also calls into question the true presence and prevalence of some of these species, and whether confusion has occurred based on host species.

It is difficult to draw significant conclusions in relation to the TP of several of the included STH species in Indigenous communities, and more contemporary data across several host species is needed to achieve a clearer TP estimate. However, both *A. caninum* and *S. stercoralis* appear to remain endemic in similar proportions to those in the past 30–40 years. Knowledge gaps remain for *A. duodenale*, *N. americanus*, *A. lumbricoides*, the role of dogs as reservoirs of infection with *S. stercoralis*, as well as the importance of the increasing population of cats as reservoirs of zoonotic STH infection. By further understanding true prevalence across host species, a clearer picture of parasite status can be achieved which can guide culturally relevant, Indigenous-led policy and effective treatment and prevention strategies. Through these holistic, One Health approaches it may be possible to diminish and eliminate these important parasites.

## Supporting information

**S1 PRISMA Checklist. PRISMA checklist.**
(DOCX)

**S1 Table. Extracted data and assessment of bias of all included papers.**
(XLSX)

**S1 File. Code for Bayesian analysis of true prevalence using WinBUGS.**
(R)

**S2 File. Code for Rogan-Gladen analysis of true prevalence.**
(R)

**S3 File. Code for forest plot with pooled true prevalence.**
(R)

**S4 File. Code for spatial plot of included studies.**
(R)

## Author Contributions

**Conceptualization:** Cameron Raw, Rebecca J. Traub, Anke Wiethoelter.

**Data curation:** Cameron Raw, Anke Wiethoelter.

**Formal analysis:** Cameron Raw, Rebecca J. Traub, Mark Stevenson, Anke Wiethoelter.

**Methodology:** Cameron Raw, Rebecca J. Traub, Anke Wiethoelter.

**Project administration:** Cameron Raw, Rebecca J. Traub, Anke Wiethoelter.

**Software:** Cameron Raw, Anke Wiethoelter.

**Supervision:** Rebecca J. Traub, Mark Stevenson, Anke Wiethoelter.

**Validation:** Cameron Raw, Patsy A. Zendejas-Heredia.

**Visualization:** Cameron Raw.

**Writing – original draft:** Cameron Raw.

**Writing – review & editing:** Cameron Raw, Rebecca J. Traub, Patsy A. Zendejas-Heredia, Mark Stevenson, Anke Wiethoelter.

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
