## [Decision Letter · Decision Letter 0]

3 Aug 2022

Dear Mr Raw,

Thank you very much for submitting your manuscript "A systematic review and meta-analysis of human and zoonotic dog soil-transmitted helminth infections in Australian Indigenous communities" for consideration at PLOS Neglected Tropical Diseases. As with all papers reviewed by the journal, your manuscript was reviewed by members of the editorial board and by several independent reviewers. The reviewers appreciated the attention to an important topic. Based on the reviews, we are likely to accept this manuscript for publication, providing that you modify the manuscript according to the review recommendations. 

Please address the points raised by all the reviewers, paying particular attention to the issues raised by Reviewers 1 and 2.

Sincerely,

Siddhartha Mahanty, M.B.B.S., M.P.H

Academic Editor

Nadira Karunaweera

Section Editor

Reviewer's Responses to Questions

**Key Review Criteria Required for Acceptance?**

**Methods**

-Are the objectives of the study clearly articulated with a clear testable hypothesis stated?

-Is the study design appropriate to address the stated objectives?

-Is the population clearly described and appropriate for the hypothesis being tested?

-Is the sample size sufficient to ensure adequate power to address the hypothesis being tested?

-Were correct statistical analysis used to support conclusions?

-Are there concerns about ethical or regulatory requirements being met?

Reviewer #1: See general comments

Reviewer #2: -Are the objectives of the study clearly articulated with a clear testable hypothesis stated? Yes

-Is the study design appropriate to address the stated objectives? Yes

-Is the population clearly described and appropriate for the hypothesis being tested? Yes

-Is the sample size sufficient to ensure adequate power to address the hypothesis being tested? Yes

-Were correct statistical analysis used to support conclusions? No - see below comments on zinc-sulfate flotation TP

-Are there concerns about ethical or regulatory requirements being met? No

Table 1 Legend: table legends should stand alone, please provide more detail in the figure legend of what this table shows.

Also, with reference to Table 1:

a) In my experience, the sensitivity of flotation techniques for Strongyloides is much lower. In fact, the reference given for the diagnostic sensitivity of zinc-sulfate flotation reports combined data of both zinc sulfate flotation and formalin -ethyl acetate flotation (one or the other). This is not a suitable reference to use for these calculations. Please find an alternative, suitable, reference and re-calculate your true prevalence data for this methodology.

b) Can you state that this is data for the commonly used Verweij PCR? There are several, each with unique sensitivity and specificity, was this the only method used in all the papers reviewed? 

c) Which ELISA serology? As above, sensitivity and specificity of ELISA kits vary by serological antigen (and has not been reported for some in-house developed kits). If within an acceptable range, you may be able to use the one value to determine TP, but this should be considered and addressed in the review.

Reviewer #3: Yes to all of these criteria.

**Results**

-Does the analysis presented match the analysis plan?

-Are the results clearly and completely presented?

-Are the figures (Tables, Images) of sufficient quality for clarity?

Reviewer #1: See general comments

Reviewer #2: -Does the analysis presented match the analysis plan? Yes

-Are the results clearly and completely presented? No - see comments below on Table S1

-Are the figures (Tables, Images) of sufficient quality for clarity? Yes

Line 245: grammar – “fitted”

Lines 267-268: Please provide the reference for this study from the Kimberley which identified A. lumbricoides infections. Perhaps also note that the Kimberley is in North-West Western Australia (for international readers who are not immediately aware of various regions of Australia)?

Line 285: What about true dingos (not hybridised)? Might any of these have been included in the study?

Line 291-292: Can you break down which serological kits were used (in-house, commercial, antigen type) and put this in a supplementary table? This has relevance to the prevalences reported – this data may be included as a column in the S1 Table.

Lines 293, 296: Ditto for conventional PCR and qPCR methodologies.

Table S1 needs a superior legend – what do the green and amber colours in some of the columns mean? What does L, M and H mean (low, medium, and high? Referring to what? For instance, columns are titled things like, “Random or census” and “length of shortest prevalence period appropriate?”. Results are then noted as “low”, “medium” or “high”– shouldn’t this be listed “random” or “census”; or as a length of shortest prevalence period (depending on the column)?? I cannot determine how to interpret some aspects of this table.

Figure 3: Please include the sampling dates that these data span for each species

Reviewer #3: This is a review, and so the results per se are the outcome of the estimation of true prevalences derived from the literature that met inclusion criteria. These results are described clearly.

**Conclusions**

-Are the conclusions supported by the data presented?

-Are the limitations of analysis clearly described?

-Do the authors discuss how these data can be helpful to advance our understanding of the topic under study?

-Is public health relevance addressed?

Reviewer #1: See general comments

Reviewer #2: -Are the conclusions supported by the data presented? Yes

-Are the limitations of analysis clearly described? Yes

-Do the authors discuss how these data can be helpful to advance our understanding of the topic under study? Yes

-Is public health relevance addressed? Yes

Lines 402-412: Just a comment for the author's information and consideration here, which the authors may wish to include in this discussion paragraph or not include (it is not essential to the text). There was once, and still is in some circles, an incorrect belief that all human hookworms in Australia were due to A. duodenale. So, all hookworm infections in humans have been assigned to this species in many papers, particularly circa 1970-2000. Prociv found Necator in Australia and two human A. ceylanicum infections have been identified, so clearly this assumption was incorrect.

Line 444: Can a more scientific term than “ill thrift” be used?

Reviewer #3: The conclusions are sound and not surprising.... the aggregation of data from studies spanning >20 years and carried out mostly with "old" techniques with limitations on specificity and sensitivity inevitably leads to conclusions hedged by caveats related to the methods. Although not surprising, it is worth pointing these limitations out, if only to promote the research agenda of the laboratory from whom this review origninates. The fact that is is somewhat self-promotional does not detract from the message that more modern methods with greater specificity in particular are required.

**Editorial and Data Presentation Modifications?**

Reviewer #1: See general comments

Reviewer #2: Abstract & Author Summary:

Lines 20-27: please make clear if these pooled prevalence rates are for dogs or humans. I assume dogs as you note A. caninum, but on initial reading, this is not clear, as later you mention T. trichiura – please differentiate the relevant host species for these results.

Line 36: can you rewrite to make clear that “zoonotic” means they cross between humans and animals?

Introduction:

Line 49 – perhaps “are Neglected Tropical Diseases” instead of “belong to”?

Line 52: “may act as zoonotic reservoirs” – it depends on the STH if they do

Line 57: please add A. ceylanicum to the list of hookworms

Line 65-67: I think this sentence needs to be rewritten somewhat as it reads as though S. stercoralis is a confirmed zoonosis from cats and dogs. This is probably, but not yet proven.

Line 84-85: It is not the motility of L3 hookworm and Strongyloides that allows them to infect their hosts percutaneously, but rather the enzymes they produce which allow skin penetration. This sentence should be reworded to avoid the misconception that motility alone is responsible for percutaneous penetration.

Line 107: “…in Australian Aboriginal communities” – please provide the context of where this One Health understanding is required due to culturally relevant bonds.

References: 

Some references are incomplete or not in PLoS NTDs format (e.g. Lima & Delgado). Please review and correct the reference title list as necessary.

Reviewer #3: The review is well written. There are a few typos here and there that will no doubt be caught in the revision.

**Summary and General Comments**

Reviewer #1: Line 11: I wonder how good have the control efforts been 

Introduction

Line 59: Ascaris is rare in Australia – has it been found in remote communities in the last few decades - so not sure if it is super relevant. 

Line 72: italics S. stercoralis

Line 95-97: reference for this sentence

Line 109: Spp. for Strongyloides as well. comma after intestines and colon

Was the meta-analysis performed on a specific question? Which was? Prevalence in humans and dogs in Australia? Should state along with the systematic review goals.

I think the introduction needs to be tightened up overall. Maybe just re-organised. Perhaps add subheadings and organise that way. I’m not sure all the information is totally necessary, and perhaps some of this information could be moved to discussion. So, a shorter introduction, go straight into methods and results, then use some of this info to put in to context. 

Methods

211: authors or independent?

Table 2: This is obviously restricted to what methods were performed in the included papers, but really no agar plate or Baermann methods were used in the included papers? Thinking specifically of strongyloides.

Strongyloides stool ID is based primarily on larvae as eggs tend to hatch in the gut – very heavy infections you may still get eggs in the stool, but lyse very quickly (as do the hookworm eggs), when DS, FE, and SSF is being performed in these studies are they doing morphological ID of the larvae, and at what stage of larvae? FE and SSF are fixatives, and presumably DS is being done immediately, obviously they are not going to be looking totally at L3 larvae. 

Line 299: what year was the Ascaris study published?

Line 426-428: Hmmm makes you really doubt if it is strongyloides in the one’s that don’t use these techniques to confirm (if they used one of the other microscopic techniques initially and not ELISA or PCR)

Line 438: was it mentioned in methods that analysis was done in two time periods for this figure?

Discussion is good. 

It won’t be relevant for inclusion in the analysis, but perhaps have a look at this paper for some wider strongy, more current (as you indicate, not much on any STH last decade met inclusion criteria), context in Australia.

Shield, J., S. Braat, M. Watts, G. Robertson, M. Beaman, J. McLeod, R. W. Baird, J. Hart, J. Robson, R. Lee, S. McKessar, S. Nicholson, J. Mayer-Coverdale and B. A. Biggs (2021). "Seropositivity and geographical distribution of Strongyloides stercoralis in Australia: A study of pathology laboratory data from 2012-2016." PLoS Negl Trop Dis 15(3): e0009160.

Reviewer #2: This paper is s systematic review of soil transmitted helminths (STH) in humans and dogs (where they are zoonotic) in Australian Aboriginal communities. Overall, a well-considered and thoughtful approach to the review process and presentation has been undertaken. Important considerations like the lack of a true gold standard for many diagnostic tests and difficulty in differentiating hookworm genera by egg appearance have been included in the discussion. 

In my opinion, this is is an important reference and addition to an area that is increasingly topical. However, there are still some elements of this manuscript which require consideration and/or improvement before it is suitable for publication, as discussed in the above review comments.

Reviewer #3: See above. This review is a timely reminder of the poor state of STH diagnostics in general, and a useful resource for work on the largely unrecognised problem of STH infection in Australian first nations people living in remote settements.

PLOS authors have the option to publish the peer review history of their article (what does this mean?). If published, this will include your full peer review and any attached files.

Reviewer #1: No

Reviewer #2: No

Reviewer #3: Yes: Warwick Grant

Figure Files:

Data Requirements:

Reproducibility:

References

---

## [Decision Letter · Decision Letter 1]

20 Sep 2022

Dear Mr Raw,

Thank you very much for submitting your manuscript "A systematic review and meta-analysis of human and zoonotic dog soil-transmitted helminth infections in Australian Indigenous communities" for consideration at PLOS Neglected Tropical Diseases. As with all papers reviewed by the journal, your manuscript was reviewed by members of the editorial board and by several independent reviewers. The reviewers appreciated the attention to an important topic. Based on the reviews, we are likely to accept this manuscript for publication, providing that you modify the manuscript according to the review recommendations. 

Please address the point of concern raised by reviewer 2 and consider the modifications suggested by the reviewers.

Sincerely,

Siddhartha Mahanty, M.B.B.S., M.P.H

Academic Editor

Nadira Karunaweera

Section Editor

Reviewer's Responses to Questions

**Key Review Criteria Required for Acceptance?**

**Methods**

-Are the objectives of the study clearly articulated with a clear testable hypothesis stated?

-Is the study design appropriate to address the stated objectives?

-Is the population clearly described and appropriate for the hypothesis being tested?

-Is the sample size sufficient to ensure adequate power to address the hypothesis being tested?

-Were correct statistical analysis used to support conclusions?

-Are there concerns about ethical or regulatory requirements being met?

Reviewer #1: (No Response)

Reviewer #2: (No Response)

**Results**

-Does the analysis presented match the analysis plan?

-Are the results clearly and completely presented?

-Are the figures (Tables, Images) of sufficient quality for clarity?

Reviewer #1: (No Response)

Reviewer #2: (No Response)

**Conclusions**

-Are the conclusions supported by the data presented?

-Are the limitations of analysis clearly described?

-Do the authors discuss how these data can be helpful to advance our understanding of the topic under study?

-Is public health relevance addressed?

Reviewer #1: (No Response)

Reviewer #2: (No Response)

**Editorial and Data Presentation Modifications?**

Reviewer #1: (No Response)

Reviewer #2: Line 110: While S. stercoralis infections are self-limiting clinically for dogs, I have not seen evidence for this in humans, please check this statement is correct for human hosts and amend if necessary

I cannot see the changes in table S1 that are described in the response to reviewers. Perhaps an old version of Table S1 has been uploaded?

**Summary and General Comments**

Reviewer #1: Line 69: space caninum[6-9]

Line 69: A. caninum

Line 87-88: Consider moving this to after the next line about third stage larvae penetrating. As my first thought on reading ‘in the same manner’ was, well mostly they penetrate the skin directly. 

Line 92: all STH or just hookworm? As Ascaris is rare the STH in Australia would have to be made up of hookworm and Trichuris infection, unless the referenced article also included Strongyloides in the STH grouping which doesn’t always happen.

Reviewer #2: (No Response)

PLOS authors have the option to publish the peer review history of their article (what does this mean?). If published, this will include your full peer review and any attached files.

Reviewer #1: Yes: Catherine Gordon

Reviewer #2: Yes: Richard Stewart Brabdbury

Figure Files:

Data Requirements:

Reproducibility:

References

---

## [Editor Report · Decision Letter 2]

14 Oct 2022

Dear Mr Raw,

We are pleased to inform you that your manuscript 'A systematic review and meta-analysis of human and zoonotic dog soil-transmitted helminth infections in Australian Indigenous communities' has been provisionally accepted for publication in PLOS Neglected Tropical Diseases.

Best regards,

Siddhartha Mahanty, M.B.B.S., M.P.H

Academic Editor

Nadira Karunaweera

Section Editor

---

## [Editor Report · Acceptance letter]

19 Oct 2022

Dear Mr Raw,

We are delighted to inform you that your manuscript, "A systematic review and meta-analysis of human and zoonotic dog soil-transmitted helminth infections in Australian Indigenous communities," has been formally accepted for publication in PLOS Neglected Tropical Diseases.

Best regards,

Shaden Kamhawi

co-Editor-in-Chief

Paul Brindley

co-Editor-in-Chief
